# Zoonotic Viruses in Three Species of Voles from Poland

**DOI:** 10.3390/ani10101820

**Published:** 2020-10-06

**Authors:** Maciej Grzybek, Katarzyna Tołkacz, Tarja Sironen, Sanna Mäki, Mohammed Alsarraf, Jolanta Behnke-Borowczyk, Beata Biernat, Joanna Nowicka, Antti Vaheri, Heikki Henttonen, Jerzy M. Behnke, Anna Bajer

**Affiliations:** 1Department of Tropical Parasitology, Institute of Maritime and Tropical Medicine, Medical University of Gdansk, Powstania Styczniowego 9B, 81-519 Gdynia, Poland; beata.biernat@gumed.edu.pl (B.B.); joanna.nowicka@gumed.edu.pl (J.N.); 2Department of Eco-Epidemiology for Parasitic Diseases, Faculty of Biology, University of Warsaw, 1 Miecznikowa Str, 02-096 Warsaw, Poland; k.tolkacz@ibb.waw.pl (K.T.); muha@biol.uw.edu.pl (M.A.); anabena@biol.uw.edu.pl (A.B.); 3Department of Antarctic Biology, Institute of Biochemistry and Biophysics, Polish Academy of Sciences, 5A Pawińskiego Str, 02-106 Warsaw, Poland; 4Department of Virology, University of Helsinki, Haartmaninkatu 3, 00014 Helsinki, Finland; tarja.sironen@helsinki.fi (T.S.); sanna.z.maki@helsinki.fi (S.M.); antti.vaheri@helsinki.fi (A.V.); 5Department of Forest Pathology, Poznan University of Life Sciences, Wojska Polskiego 71c, 60-625 Poznan, Poland; jbehnke@up.poznan.pl; 6Natural Resources Institute Finland, Latokartanonkaari 9, 00790 Helsinki, Finland; ext.Heikki.Henttonen@luke.fi; 7School of Life Sciences, University of Nottingham, University Park, Nottingham NG7 2RD, UK; jerzy.behnke@nottingham.ac.uk

**Keywords:** arenavirus, hantavirus, LCMV, CPXV, PUUV, TULV, microtus, Alexandromys, vole, seroprevalence, Poland

## Abstract

**Simple Summary:**

Wild rodents constitute a significant threat to public health. We tested 77 voles from northeastern Poland for the presence of antibodies to hantaviruses, arenaviruses and cowpox viruses. We report 18.2% overall seroprevalence of zoonotic viruses. Our results contribute to knowledge about the role of Polish voles as possible reservoirs of viral infections.

**Abstract:**

Rodents are known to be reservoir hosts for a plethora of zoonotic viruses and therefore play a significant role in the dissemination of these pathogens. We trapped three vole species (*Microtus arvalis, Alexandromys oeconomus* and *Microtus agrestis*) in northeastern Poland, all of which are widely distributed species in Europe. Using immunofluorescence assays, we assessed serum samples for the presence of antibodies to hantaviruses, arenaviruses and cowpox viruses (CPXV). We detected antibodies against CPXV and Puumala hantavirus (PUUV), the overall seroprevalence of combined viral infections being 18.2% [10.5–29.3] and mostly attributed to CPXV. We detected only one PUUV/TULV cross-reaction in *Microtus arvalis* (1.3% [0.1–7.9]), but found similar levels of antibodies against CPXV in all three vole species. There were no significant differences in seroprevalence of CPXV among host species and age categories, nor between the sexes. These results contribute to our understanding of the distribution and abundance of CPXV in voles in Europe, and confirm that CPXV circulates also in *Microtus* and *Alexandromys* voles in northeastern Poland.

## 1. Introduction

The identification of possible hosts, and the study of transmission dynamics in their populations, are both crucial steps in controlling zoonotic diseases [1,2]. Rodents, the most widespread and abundant mammals, are considered to be a significant source of zoonotic pathogens [1,3]. Studies of the population dynamics of wild rodents have demonstrated that European rodent populations experience multiannual and cyclic density fluctuations [4,5]. This has been linked with variation in the incidence of zoonoses spread by rodents [4,5].

Rodent-borne hanta-, mammarena- and orthopox viral pathogens are maintained in nature by direct intraspecies, or, as probably is the case with cowpox virus (CPXV), interspecies transmission from rodent to rodent without the participation of arthropod vectors. Transmission among rodents occurs by contact with body fluids or excretions [6,7,8,9].

The most prevalent rodent-borne viruses carried by European rodents include hantaviruses, the Mammarena lymphocytic choriomeningitis virus (LCMV), and the Orthopox CPXV [10,11,12,13,14,15]. The Puumala hantavirus (PUUV) is widely prevalent in bank vole (*Myodes glareolus*) populations [7,16]. Hantavirus infections in bank voles are chronic and therefore viral replication and shedding are persistent [9,17]. As a consequence, after being infected, the rodent host may be infectious for the rest of its lifespan [18].

CPXV is the only known wildlife-borne orthopox virus (OPV) in Europe [11]. Field voles (*Microtus agrestis)*, among other rodent species (i.e., *Apodemus sylvaticus* and *Myodes glareolus*), are known to act as reservoir hosts for CPXV [11,19,20]. LCMV, the only arenavirus in Europe, was thought to be prevalent in the house mouse (*Mus musculus*) [21]. However, further studies have revealed that the virus is present also in other murine and vole species [21,22,23,24]. Ledesma et al. [25] identified an independent genetic lineage of LCMV in wood mice and this led to the suggestion of spillover and the circulation of multiple related and cross-reactive Mammarena viruses.

In 2018, 29 European countries reported 1826 cases of hantavirus disease, mainly caused by Puumala virus [26]. The first outbreak of hantavirus infections in Poland was reported in 2007 when nine human cases were diagnosed, and was followed by a further 93 cases in the period 2014–2018 [26]. One human cowpox infection was reported in Poland in 2015 [27].

Voles (*Microtus* spp., *Myodes* spp. and *Alexandromys* spp.) are the most abundant rodents in European grasslands and forests [28]. This study aimed to evaluate zoonotic and potentially zoonotic viruses in populations of Microtus and Alexandromys spp. in northeastern Poland. Our results contribute to the understanding of the role of vole populations in the maintenance and dissemination of viral pathogens in this geographical region.

## 2. Materials and Methods

### 2.1. Ethical Approval

This study was carried out with due regard for the principles required by the European Union and the Polish Law on Animal Protection. Formal permits were obtained, allowing trapping in the field and subsequent laboratory analysis of sampled materials. Our project was approved by the First Warsaw Local Ethics Committee for Animal Experimentation (ethical license numbers: 148/2011 and 406/2013).

### 2.2. Collection of Voles

The study site was located in the Mazury Lake District region in the northeastern corner of Poland (Urwitałt, near Mikołajki; 53°48’50.25”N, 21°39’7.17”E) and previously described [29,30]. Voles were collected in August 2013 during the late summer season, when rodent population density is at its highest in the annual cycle. Voles were live-trapped using mixed bait comprising fruit (apple), vegetables (carrot and cucumber), and grain. Two traps were set every 10 m along the trap lines at dusk. The following morning traps were checked and closed to prevent animals from entering during daytime and to avoid losses from excessive heat from exposure of traps to direct sunlight. Traps were then re-baited and reset on the following afternoon. All traps were also closed during periods of intensive rainfall. All captured voles were transported in their traps to the laboratory for inspection.

The autopsies were carried out under terminal isoflurane anaesthesia. Animals were weighed to the nearest gram, total body length, and tail length were measured in millimetres. Animals were allocated to three age classes (juveniles, subadult, and adults), based on body weight and nose-to-anus length together with a reproductive condition (scrotal, semi-scrotal, or non-scrotal for males; lactating, pregnant or receptive for females) [29,30,31,32]. Recent reports suggest taxonomic changes within vole species [33]. Here, we refer to *Alexandromys oeconomus* (=*Microtus eoconomus*) following Lissovsky et al. [34] and Zorentko et al. [35]. We confirmed species identity by examination of the lower molars M_1_ and M_2_ and the second upper molar M^2^, especially to distinguish between juvenile individuals of *A. oeconomus* and *M. agrestis* [36].

Blood samples were collected directly from the heart using a sterile 1.5 mL syringe immediately after death from over-exposure to Isoflurane (Baxter, Deerfield, IL, USA) anaesthetic. Samples were centrifuged at 5000 rpm for 10 min. Serum was collected and stored at −80 °C until the samples could be analyzed on completion of the fieldwork.

### 2.3. Serological Screening ofAnti-Virus Antibodies

Serum samples were analyzed using an immunofluorescence assay (IFA). The serum samples were diluted 1:10 in PBS and the reactivity of the samples to hantaviruses was tested with PUUV-(Puumala virus)-IFA, to cowpox viruses with CPXV (Cowpox virus)-IFA and arenaviruses with LCMV (Lymphocytic choriomeningitis virus)-IFA. PUUV (Sotkamo strain), CPXV, and LCMV (Armstrong strain)-infected Vero E6 cells were detached with trypsin, mixed with uninfected Vero E6 cells (in a ratio of 1:3), washed with PBS, spotted on IFA slides, air-dried, and fixed with acetone as described earlier [37]. The slides were stored at −70 °C until use. TULA orthohantavirus (TULV), specific for *Microtus* voles, is known to cross-react strongly with PUUV antibodies (and vice versa). Thus, we report it as PUUV/TULV seroprevalence.

IFAs were carried out as previously described [38] with seropositive human serum as a positive control for the PUUV- and CPXV-IFA; and LCMV mouse monoclonal antibody (Progen, Heidelberg, Germany) for the LCMV-IFA. The slides were read under a fluorescence microscope and photographs were taken with a ZOE^TM^ fluorescent cell imager (BioRad, Hercules, CA, USA).

### 2.4. Statistical Analysis

Prevalence values (percentage of animals infected) are given with 95% confidence limits in parenthesis (CL_95_) or error bars on figures, calculated by bespoke software based on the tables of Rohlf and Sokal [39].

The statistical approach has been documented comprehensively in our earlier publications [40,41,42,43,44]. For analysis of prevalence, we used maximum likelihood techniques based on log-linear analysis of contingency tables in the software package IBM SPSS Statistics Version 21 (IBM Corporation, Armonk, NY, USA).

## 3. Results

We screened a total of 77 *Microtus* and *Alexandromys* spp. serum samples for the presence of mammarena-, orthopox- and hantavirus antibodies. We confirmed the presence of antibodies against CPXV and PUUV/TULV. No individuals were seropositive for LCMV. 

The overall seroprevalence of zoonotic viruses was 18.2% (10.5–29.3). Most of the seropositivity was for CPXV (16.9% [9.4–27.9]) with only one individual (*M. arvalis*) showing evidence of the presence of anti-hantaviral antibodies (1.3% [0.1–7.9]). Further analysis is therefore confined to the seroprevalence of CPXV.

CPXV antibodies were present in all three vole species. However, there were no significant differences between vole species (Figure 1A). Female voles had a marginally higher value for CPXV seroprevalence than males (18.2% [8.4–34.8] and 15.2% [7.0–28.7], respectively) (Figure 1B), but the difference between the sexes was not significant. There was no significant effect of host age on CPXV seroprevalence (Figure 1C).

## 4. Discussion

We report a high overall seroprevalence of two common viral infections, mostly CPXV (18.2%) in voles from northeastern Poland. These findings are not only of considerable relevance to public health in the region but by inference are likely to have relevance also for other European regions populated by *Microtus* spp. High seroprevalence of CPXV is consistent with our report on seroprevalence of zoonotic viruses in bank voles (*M. glareolus*) from the same geographical region [14]. It is also in agreement with those obtained in different parts of Europe (e.g., Finland, England, and Turkey) where CPXV, PUUV, and TULV virus species have been detected in voles [22,45,46,47,48,49,50,51]. We did not detect any seropositivity to LCMV in the vole species we sampled, although LCMV has been found in other rodent populations (both mice and vole species) in different parts of Europe [21,22,23,38].

The current work was based on the presence/absence of specific antibody against viruses, and hence positivity in our assay reflected the history of previous infections and not necessarily current infection [41]. Serological tests may provide false positive results due to crossreactivity. This may be caused by the circulation of multiple related and cross-reactive arenaviruses and hantaviruses in rodents [52]. The “gold standard” for detection of zoonotic viruses in rodents should include serological and molecular approaches. Searching for antibodies followed by immunoblotting of spleen/lung tissues, and PCR/RT-PCR for detection of viral DNA/RNA may be applied [53,54]. However, this requires adequate types of samples (host tissues: spleen, lungs, brain), preservation methods (−80 °C), reagents (for RNA preservation), and increased field and laboratory workload [55]. Nevertheless, all these methods may still have limitations because of virus biology. For example, CPXV is a DNA virus with viremia lasting for 2–3 weeks, whereas hantaviruses and arenaviruses cause chronic infection. Therefore, the transmission dynamics of these viruses differ [47]. For the same reason, PCR identification of CPXV-positive individuals can create similar difficulties [11].

Intrinsic factors such as host age, maturity, and host sex may all influence the host’s exposure and susceptibility to viral infections [14,40]. On this basis, we would have expected to find a higher seroprevalence among the older animals, which would have had more opportunity for exposure to, and hence experience of infection than juveniles, but the difference between the age classes in the current study was not significant.

In summary, we found serological evidence for the presence of CPXV in three vole species, and PUUV/TULV in a single *M. arvalis* vole, from wild vole populations in northeastern Poland. We believe that identifying rodent species that can serve as reservoirs of zoonotic diseases and predicting regions where new outbreaks are most likely to happen are crucial steps in preventing and minimising the extent of zoonotic diseases in humans [56]. Our results help to consolidate the gap in knowledge about the role of Polish voles as possible candidates for reservoirs of viral infections.

## Figures and Tables

**Figure 1 animals-10-01820-f001:**
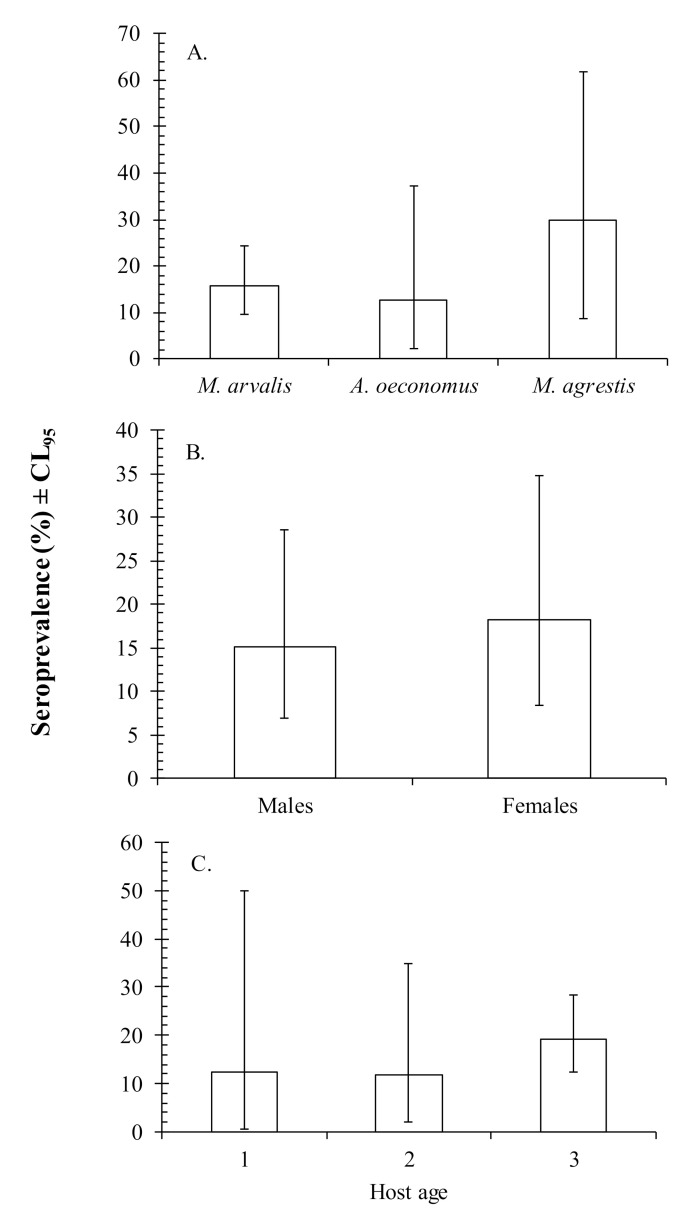
Seroprevalence of cowpox virus within: (**A**) three vole species; (**B**) host sex; (**C**) host age. Number of animals sampled: *Microtus arvalis* (*n* = 51); *Alexandromys oeconomus* (*n* = 16); *Microtus agrestis* (*n* = 10).

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
