# Peer review of "Zoonotic Viruses in Three Species of Voles from Poland"

_animals, 2020, doi:10.3390/ani10101820_

Round 1
Reviewer 1 Report
This interesting manuscript describes the seroprevalence of wild mammals for some zoonotic virus in Poland. I suggest some revisions to have it published.
Simple summary
-line 32-3: I would change in: ‘We tested 32 voles from northeastern Poland for the presence of antibodies to hantaviruses, arenaviruses and cowpox viruses’
Please change ‘NE’ in ‘northeastern’ across the manuscript text.
-line 35-26: you surely do not want to ‘consolidate a gap’, you rather want to fill it.. I would change in:’ Our results contribute to the knowledge about the role of Polish voles as possible reservoirs..’
Abstract:
-line 42: I would change in:’ …distributed species in Europe. By immunofluorescence assays, we tested serum samples for the presence of antibodies to..’
-line 46-48: I would omit this sentence on cross-reaction from the abstract.
-lines 49 onwards – I would change in: ‘we found 48 antibodies against CPXV in all three vole species. There were no significant differences in seroprevalence among species and host age categories, nor between the sexes. Our results contribute to the understanding of CPXV distribution and abundance in voles in Europe, and confirm that CPXV circulates also in Microtus and Alexandromys voles in northeastern Poland.’
Introduction:
-lines 65-6: please change in ‘Rodents, the most widespread and abundant mammals, are considered to be a significant source of zoonotic pathogens’
-line 70: delete ‘of’
-lines 72-3: I would move this sentence below, where you are talking about the specific pathogens (indeed, you say again in line 78, that hantavirus cause a persistent infection.
-line 74: transmission ‘among’ rodents instead of ‘in’
-line 89-90: the sentence is in italic
-Are these viral diseases notified in Poland? Could you add here a sentence here about their impact on human health (if known) or absence of data (if not known..)?
-lines 93-4: this last sentence could be omitted, on my opinion.
M&M
-what was your trapping effort? (number of night/traps?)
-line 145: ‘figures’, not ‘figs’
Results
-I would delete the adverb ‘however’ in lines 157 and 161.
Discussion
-line 184: delete ‘and’
-line 189: as far as I understand, cross-reacting PUUV antigen was used for Hantavirus serology, not for CPXV.. please revise the sentence.
-It would be important in the discussion to provide more details on studies carried out in other countries and mammal species, in order to compare your results and justify your statement that you have a ‘high overall prevalence’. Please also better explain what you mean in lines 179-80 with ‘but see Forbes et al.’
Author Response
We provided a point by point response to each of the issues raised, indicating how we have modified the ms, and explaining our position when we felt it inappropriate to make changes.
Reviewer 1
This interesting manuscript describes the seroprevalence of wild mammals for some zoonotic virus in Poland. I suggest some revisions to have it published.
*OUR RESPONSE: We thank Reviewer 1 for positive comments concerning our manuscript.
Simple summary
-line 32-3: I would change in: ‘We tested 32 voles from northeastern Poland for the presence of antibodies to hantaviruses, arenaviruses and cowpox viruses’
Please change ‘NE’ in ‘northeastern’ across the manuscript text.
-line 35-26: you surely do not want to ‘consolidate a gap’, you rather want to fill it.. I would change in:’ Our results contribute to the knowledge about the role of Polish voles as possible reservoirs..’
*OUR RESPONSE: We amended MS as suggested.
Abstract:
-line 42: I would change in:’ …distributed species in Europe. By immunofluorescence assays, we tested serum samples for the presence of antibodies to..’
-line 46-48: I would omit this sentence on cross-reaction from the abstract.
-lines 49 onwards – I would change in: ‘we found 48 antibodies against CPXV in all three vole species. There were no significant differences in seroprevalence among species and host age categories, nor between the sexes. Our results contribute to the understanding of CPXV distribution and abundance in voles in Europe, and confirm that CPXV circulates also in Microtus and Alexandromys voles in northeastern Poland.’
*OUR RESPONSE: We amended Abstract as suggested above.
Introduction:
-lines 65-6: please change in ‘Rodents, the most widespread and abundant mammals, are considered to be a significant source of zoonotic pathogens’
-line 70: delete ‘of’
-lines 72-3: I would move this sentence below, where you are talking about the specific pathogens (indeed, you say again in line 78, that hantavirus cause a persistent infection.
-line 74: transmission ‘among’ rodents instead of ‘in’
-line 89-90: the sentence is in italic
*OUR RESPONSE: We amended Abstract as suggested above.
-Are these viral diseases notified in Poland? Could you add here a sentence here about their impact on human health (if known) or absence of data (if not known..)?
*OUR RESPONSE: We provided epidemiological data from ECDC.
-lines 93-4: this last sentence could be omitted, on my opinion.
*OUR RESPONSE: We deleted this sentence.
M&M
-what was your trapping effort? (number of night/traps?)
*OUR RESPONSE: Our trapping effort was 5.7% within 1616 trap nights.
-line 145: ‘figures’, not ‘figs’
Results
-I would delete the adverb ‘however’ in lines 157 and 161.
*OUR RESPONSE: We amended the ms suggested above.
Discussion
-line 184: delete ‘and’
-line 189: as far as I understand, cross-reacting PUUV antigen was used for Hantavirus serology, not for CPXV.. please revise the sentence.
-It would be important in the discussion to provide more details on studies carried out in other countries and mammal species, in order to compare your results and justify your statement that you have a ‘high overall prevalence’. Please also better explain what you mean in lines 179-80 with ‘but see Forbes et al.’
*OUR RESPONSE: We amended the ms according to your suggestions.
Reviewer 2 Report
The authors postulated that zoonotic pathogens, specifically hantaviruses, arenaviruses and pox viruses were represented in three vole species found in the north-east region of Poland. This paper extends and confirms the author’s previous work. While limited in scope, this paper is well written and provides a valuable benchmark for these three viruses within these three vole species in this region. I only have a few minor points to make
Line 89 unnecessary italics for text.
Line 119 further explanation - this is probably according to 32 33
Line 172 the first sentence is contentious, as it generalizes 3 viral infections to all viral infections. This should be more precisely phrased, for example “…seroprevalence of 3 common zoonotic viral infections, mostly ….”.
Author Response
We provided a point by point response to each of the issues raised, indicating how we have modified the ms, and explaining our position when we felt it inappropriate to make changes.
Reviewer 2
The authors postulated that zoonotic pathogens, specifically hantaviruses, arenaviruses and pox viruses were represented in three vole species found in the north-east region of Poland. This paper extends and confirms the author’s previous work. While limited in scope, this paper is well written and provides a valuable benchmark for these three viruses within these three vole species in this region. I only have a few minor points to make
*OUR RESPONSE: We thank Reviewer 2 for positive comments concerning our manuscript.
Line 89 unnecessary italics for text.
*OUR RESPONSE: MS amended as suggested.
Line 119 further explanation - this is probably according to 32 33
*OUR RESPONSE: We clarified the text. Taxonomical change has been mentioned and proper references have been added.
Line 172 the first sentence is contentious, as it generalizes 3 viral infections to all viral infections. This should be more precisely phrased, for example “…seroprevalence of 3 common zoonotic viral infections, mostly ….”.
*OUR RESPONSE: We agree. We amended MS as suggested.